# Crowd-sourced observations of a polyphagous moth reveal evidence of allochronic speciation varying along a latitudinal gradient

**Emily N. Black**, **Jarrett D. Blair**, **Karin R. L. van der Burg**, **Katie E. Marshall** *

Department of Zoology, University of British Columbia, Vancouver, British Columbia, Canada

\* kmarshall@zoology.ubc.ca

## Abstract

Allochronic speciation, where reproductive isolation between populations of a species is facilitated by a difference in reproductive timing, depends on abiotic factors such as seasonality and biotic factors such as diapause intensity. These factors are strongly influenced by latitudinal trends in climate, so we hypothesized that there is a relationship between latitude and divergence among populations separated by life history timing. *Hyphantria cunea* (the fall webworm), a lepidopteran defoliator with red and black colour morphs, is hypothesized to be experiencing an incipient allochronic speciation. However, given their broad geographic range, the strength of allochronic speciation may vary across latitude. We annotated >11,000 crowd-sourced observations of fall webworm to model geographic distribution, phenology, and differences in colour phenotype between morphs across North America. We found that red and black morph life history timing differs across North America, and the phenology of morphs diverges more in warmer climates at lower latitudes. We also found some evidence that the colour phenotype of morphs also diverges at lower latitudes, suggesting reduced gene flow between colour morphs. Our results demonstrate that seasonality in lower latitudes may increase the strength of allochronic speciation in insects, and that the strength of sympatric speciation can vary along a latitudinal gradient. This has implications for our understanding of broad-scale speciation events and trends in global biodiversity.

## Introduction

Speciation occurs when barriers to interbreeding evolve between members of a species. This can occur in two contrasting ways: allopatric speciation, where new species arise due to geographic isolation between populations, or sympatric speciation, where new species arise without geographic isolation [1]. Sympatric speciation often has an ecological component leading to reproductive isolation such as differences in abiotic tolerance, host plant use, or other changes to the niche of populations [2]. However, the presence of one of these ecological shifts does not necessitate a speciation event, as the magnitude of the ecological differences can

**Data Availability Statement:** All code, raw data underlying the results, and full-sized figures are available from the OSF Repository: https://osf.io/x64gp/.

**Funding:** E.N.B: Natural Sciences and Engineering Research Council of Canada, Canada Graduate Scholarships – Master's program (https://www.nserc-crsng.gc.ca/students-etudiants/pg-cs/cgsm-bescm_eng.asp) J.D.B.: Natural Sciences and Engineering Research Council of Canada, Postgraduate Scholarships – Doctoral program (https://www.nserc-crsng.gc.ca/students-etudiants/pg-cs/bellandpostgrad-belletsuperieures_eng.asp) K.v.d.B: National Science Foundation Integrative Organismal Systems (IOS), Grant 1930829 (https://www.nsf.gov/div/index.jsp?div=IOS) K.E.M: Natural Sciences and Engineering Research Council of Canada, Discovery Grant (https://www.nserc-crsng.gc.ca/professors-professeurs/grants-subs/dgigp-psigp_eng.asp) The funders had no role in study design, data collection and analysis, decision to publish, or preparation of the manuscript.

**Competing interests:** The authors have declared that no competing interests exist.

determine the extent of divergence [3, 4]. These ecological shifts are not static and often depend on abiotic and biotic factors that vary in space and time.

Sympatric speciation can also occur due to offsets in life history timing between populations, formally known as allochronic speciation ("allochronic" refers to the ecological separation of populations in time). These offsets may lead to the non-overlap of reproductive forms, or the formation of hybrids between populations that occupy a temporal space resulting in lower fitness (e.g., larvae that do not enter diapause in time for winter), which reinforces divergence [5, 6]. Allochronic speciation may interact with others forms of speciation; for example, it may maintain genetic isolation of formerly allopatric populations experiencing secondary contact [6, 7], evolve more easily in populations whose phenotypes are already partially diverged [8, 9], or occur in tandem with other types of ecological speciation, such as host plant specialization [10]. Allochronic speciation occurs in a variety of taxa, including insects [5, 11, 12]. For example, *Rhagoletis pomonella* (the apple maggot fly) experienced a host-plant shift with the introduction of apple trees to North America, which fruited earlier than the native hawthorn. This non-overlap of host plant phenology imposed a temporal reproductive barrier between *R. pomonella* host plant races, leading to allochronic speciation (however, this speciation may have been facilitated by allopatry; see [8, 9]) [10]. Other studies indicate that a change in voltinism (or number of generations per year) in an insect population can facilitate divergence, such as in the European corn borer (*Ostrinia nubilalis*) [13–15]. Phytophagous insects have been suggested as good candidates to study speciation and the factors contributing to its occurrence as the exceptional diversity of this group has been attributed to a high incidence of sympatric speciation [16, 17].

Though allochronic speciation occurs in temporally separated populations that exist together geographically, the progress of this speciation can still be impacted by geographic or latitudinal effects. One of the most general rules in biogeography is that biodiversity is higher in tropical regions for most taxa, possibly due to greater niche specialization and higher rate of molecular evolution in warmer climates [16, 18, 19]. As a result, Mittelbach et al. [16] postulates that divergence among populations of a species should also parallel the latitudinal gradient of biodiversity. Allochronic speciation may similarly be more effective at facilitating and maintaining population divergence in lower latitudes, as warmer climates lead to greater temporal niche space. Temporal niche space is defined as the length of time in which abiotic and biotic conditions are favourable for growth and reproduction, which is closely tied to seasonality (or cyclical changes in climate within a year [20]). Competition within the temporal niche facilitates diversification of life history patterns; however, the degree of diversification is limited by the size of the niche and the rate of evolution. When the phenological niche is larger (such as at lower latitudes), less overlap between the phenology of competing populations is expected, and vice versa [21]. Given that seasonality at lower latitudes facilitates longer growth and reproductive seasons, this suggests there is greater temporal niche space available for insects in these regions, allowing for less overlap in phenology between competing populations. This may facilitate and maintain greater diversification among populations undergoing allochronic speciation, and increase the strength of speciation at lower latitudes.

By contrast, there is evidence that speciation rates may increase at higher latitudes in particular situations. In a review of 309 bird and mammal sister species, Weir & Schluter [22] found a significant increase in speciation and extinction rates towards the poles, and North American insect species richness may increase at middle latitudes [23]. Differences in diapause initiation and termination are strong drivers of allochronic speciation in insects, and diapause timing and physiology is expected to be more important in temperate regions experiencing more extreme winter seasonality [5, 24]. This may increase the strength of allochronic speciation at higher latitudes, especially in insects who undergo facultative diapause and have greater

plasticity in life history timing [25]. This leads to two contradictory possibilities: allochronic speciation may be more effective at lower latitudes because of greater temporal niche availability, or it may be more effective at higher latitudes due to the greater degree of climate asynchrony and diapause intensity. Here, we use the fall webworm (*Hyphantria cunea*, Lepidoptera: Erebidae*)* to investigate these contrasting hypotheses, and we explore how allochronic speciation proceeds along a latitudinal gradient.

The fall webworm is a widespread generalist moth defoliator native to North America, whose gregarious larvae live in conspicuous webs on the branches of bushes and trees [26, 27]. The fall webworm appears in discrete generations, ranging from one generation per year (univoltine) in northern regions to up to four or five generations per year (multivoltine) in southern regions [6]. Fall webworm larvae occur in two distinct colour morphs in its eastern range: a red morph and a black morph which exist sympatrically [28]. The red morph has a distinct red head capsule with a brown, tan, or black body, and the black morph has a black head capsule with a grey or black body. An intermediate maroon morph exists but is only found in northwestern North America. Little is known about this colour morph's phylogenetic relatedness to the red and black morphs, but it is likely geographically separated from the other morphs by the Rocky Mountains [6].

The red and black morphs have both physiological and behavioural differences [29], and genetic analyses show clear separation of the two colour morphs, suggesting reproductive isolation with little gene flow between them despite significant range overlap [6, 26, 28, 30]. However, the two morphs are sexually compatible and produce viable F1 and F2 offspring in the lab [31, 32]. The initial divergence of the two populations was traced to 1.2–1.6 Mya, suggesting the two morphs may have initially diverged due to allopatric speciation as a result of historical glaciation or other geographic barriers [6, 33]. However, the ability of current populations to produce viable offspring when bred experimentally yet remain distinct in the wild suggests the existence of a barrier maintaining separation of the morphs. These results suggest that the two fall webworm colour morphs are currently undergoing a sympatric speciation event that is facilitating and maintaining divergence among the populations.

The black and red morph moth flights do not occur together, their voltinism differs, and the critical photoperiod for pupal diapause is longer in the black morph form, indicating that allochronic speciation may be responsible for the morph's current reproductive isolation [6, 34]. Since the fall webworm is extremely polyphagous [26] and the colour morphs share many of the same host plants [29], changes in host plant preference are an unlikely cause for this speciation event. Yang et al. [6] concluded current allochronic speciation of the fall webworm is driven by a difference in diapause intensity, development rate, and voltinism between morphs, which evolved to reduce intraspecific competition. However, they also find significant variation in voltinism and diapause timing in both morphs across their latitudinal range [6]. Given there are limited genetic analyses investigating the degree of difference between red and black fall webworm morphs across latitude [6, 30], differences in phenology between morphs can serve as an indicator of the strength of allochronic speciation acting on fall webworm across their range.

Given the fall webworm's broad distribution, crowd-sourced (i.e. citizen) science websites such as iNaturalist offer a useful alternative to field studies when investigating life history trends across a large geographic range. These websites (e.g., iNaturalist, eBird, eButterfly) are applications where users upload records or photographs of organisms they observe, which can then be annotated, sorted, and downloaded by researchers. Not only do crowd-sourced science sites reduce the cost and time required for field studies, but they produce large and high-quality datasets which engage the public in academic research [35]. iNaturalist has been used to study differences in phenology [36] and variation in colour phenotype on a large geographic

scale [37, 38], and can be used to investigate the same phenomena in the fall webworm: a conspicuous caterpillar with thousands of iNaturalist records across its range. Although crowd-sourced records are not as rigorous as field samples, their geographic, temporal, and taxonomic scope make them suitable resources for macroecological studies.

Here we use crowd-sourced iNaturalist records to investigate whether the strength of allochronic speciation varies across a latitudinal gradient using the fall webworm (*Hyphantria cunea*) as a model species. The fall webworm is found over a broad latitudinal range, and is conspicuous due to its large nests, so it is frequently photographed by community scientists. The fall webworm has two divergent populations which are facilitated and maintained by allochronic speciation [6]. It is unlikely that host plant specialization can explain the divergence between the colour morphs as the larvae are extremely polyphagous [30, 39]. We tested two competing hypotheses: either that allochronic speciation facilitates greater differences between groups at higher latitudes due to greater influence of diapause intensity and climate asynchrony, or allochronic speciation facilitates greater differences between groups at lower latitudes where there is greater temporal niche space. Due to an absence of genetic data, we used phenological overlap and phenotype measures to evaluate the strength of speciation across the fall webworm range. We find that there is less phenological overlap between red and black morph larvae at lower latitudes and in warmer climates. We also find that the colour phenotype of red and black morph larvae diverges more at lower latitudes, supporting a decrease in gene flow facilitated by allochronic speciation.

## Methods

### Data collection and standardization

We decided to focus on the fall webworm larval life stage because colour morph is only reliably discernable from larval head capsule, not adult colouration [6]. All data for geographic, phenology, and phenotype measures was sourced from images uploaded to iNaturalist (http://inaturalist.org/), a repository of community-collected observations. We found 24,991 observations of *H. cunea* across all years and locations, of which 19,333 were "research grade" (having two matching independent identifications by users) [36]. Using research grade observations helps prevent the inclusion of species commonly misidentified as fall webworm (e.g., *Malacosoma americanum*, *Estigmene acrea)* [40, 41].

We examined all research grade images of *H. cunea* uploaded from 2018–2020 because previous years had too little data to be suitable for analysis, and 2021 records were not completed at the time of analysis. We reviewed each photo manually, blinded to observation location, to identify the larval colour morph and whether the larval head capsule was visible (as head capsule is the most reliable indicator of colour morph). In total, we reviewed 11,674 iNaturalist observations. We used iNaturalist's built-in feature "Observation Fields" to store information with the photograph being classified such as whether the head capsule was visible ("Head capsule visible?") and colour morph ("Colour"). The intermediate maroon morph was classified as red in the "Colour" field [6]. No colour morph can be reliably discerned from photos of nests or moths, so we did not include these photographs in categorization or analysis.

All analyses were performed using R version 4.2.2 [42]. The dataset, all associated code, and full-sized figures are published on the Open Science Framework (https://osf.io/x64gp/). Raw data from iNaturalist includes observations from North America (which excludes invasive ranges in Asia and Europe [6, 43]), and contains the fields: photo ID, user ID, date and time of observation, observation URL, longitude and latitude, state, county, country, scientific name, and the custom Observation Fields "Head Capsule Visible?" and "Colour".

Each fall webworm observation was also assigned to their corresponding plant hardiness zone. Plant hardiness zones are geographic regions binned by their average annual minimum temperature and assigned a reference number (1–13, with lower numbers indicating colder climates) which act as an indicator of overall climatic conditions [44, 45]. Because phenology is affected by climate [46], plant hardiness zones serve as a method to bin fall webworm into groups that should experience similar seasonality, and thus, similar life history timing. Plant hardiness zones also scale roughly with latitude, so they can act as an indicator for trends in seasonality and phenology across latitudes. A plant hardiness zone map for the United States was obtained from the United States Department of Agriculture [45]. To generate a map of plant hardiness zones for Canada using the same categorization as the USA, BioSIM climate generation software was used to create a map of the average annual minimum temperature across Canada [47]. This map was then binned into plant hardiness zones based on USA categories.

Using the *tidyverse* package [48], fall webworm observations were filtered to those with an assigned colour morph and visible head capsule for all analyses. Only larval populations from the eastern United States and Canada (-100 to 0˚W longitude) were used in analyses so geographically isolated populations like those in Colorado [30] and northwestern North America were excluded. After filtering, 5,176 observations remained and were used for analysis.

## Geography

Maps of North American fall webworm distribution from 2018–2020 were generated using the *Tmap* package [49]. To determine if the colour morphs were geographically separated from one another in their eastern range, we compared the latitudinal, longitudinal, and elevational (calculated using the *Elevatr* package [50]) distribution of each colour morph in each year. Differing abiotic tolerance may cause the morphs to have different geographic distributions; however, we still expect them to be found together across their range, consistent with a sympatric form of speciation. Two-way ANOVA was used to determine if colour morph and year recorded had a significant effect on latitude, longitude, and/or elevation, and Tukey post-hoc tests were used to compare among groups. Not all observations could be assigned a valid elevation, so sample size was reduced to 4,652 observations when comparing elevation of colour morphs.

## Phenology

In addition to binning fall webworm observations by plant hardiness zone, we also binned observations by year, as phenology may change even in the same plant hardiness zone as weather varies from year to year [51, 52]. Morph abundance was calculated by week rather than by day to minimize daily variation in sampling effort by community scientists [53]. Given the red morph has a more limited latitudinal distribution than the black morph, we did not include plant hardiness zones containing less than 10 observations of the red morph in any given year in the analysis. This reduced our dataset to 4,866 observations in plant hardiness zones 5–9 for phenology measures.

Although the fall webworm appears in discrete generations each summer [6], these are difficult to parse and compare using crowd-sourced data. So, we investigated differences in phenology using two separate methods. First, we compared the earliest and latest 10% of observation dates for each colour morph. Using the earliest and latest 10% of observation dates ensures comparisons between morphs are reflective of overall larval population timing rather than individual outliers. If divergence between morphs increases with increasing temporal niche space, we expect differences between the earliest and/or latest observation dates of the

morphs will increase in higher plant hardiness zones at lower latitudes with longer growing seasons. Alternatively, if divergence between the morphs increases at higher latitudes, we expect differences between observation dates of the morphs will decrease in higher plant hardiness zones at lower latitudes. Therefore, we selected the earliest and latest 10% of observation dates for the black and red morph in each plant hardiness zone and year, and calculated the overall mean difference between red and black morph timing and the standard error of the mean difference. We then used a 3-way ANOVA and general linear model to test whether the black and red morph showed a difference in timing of their earliest and latest observances, and whether this was driven by year and plant hardiness zone.

Secondly, we used the "DTS" statistic from the *twosamples* package to quantify the degree of phenological overlap between colour morphs each year [54, 55]. The DTS statistic calculates the difference between the distributions of two groups using the weighted area between their cumulative distribution functions (aka CDFs; S2 Fig). Here we used it to quantify the degree of phenological difference between red and black morph larvae using the cumulative distribution function of their observations in each year. For each year and plant hardiness zone, we calculated the DTS statistic comparing the red and black morphs. If divergence between morphs increases with increasing temporal niche space, we would expect to see the DTS statistic increase in higher plant hardiness zones at lower latitudes. Alternatively, if divergence increases at higher latitudes, we expect to see the DTS statistic decreasing with increasing plant hardiness zone. Therefore, we used a general linear model to test whether plant hardiness zone, year, or their interaction significantly predicted DTS statistic. We compared models with both polynomial and linear predictors as initial plotting indicated the relationship between DTS statistic and plant hardiness zone may be curvilinear and selected best fit models by AIC score [56]. Since the fall webworm reproductive form (moths) does not co-occur with larvae, differences in phenological overlap of fall webworm larval generations is a predictor of differences in moth generations [26].

## Colour phenotype

We examined the colour data from iNaturalist images to determine if there is phenotypic divergence between the black and red colour morphs across plant hardiness zones. Hattori and Ito [31] found red and black morph hybrids possess an intermediate colour phenotype. Thus, in the absence of genetic data on a comparable latitudinal scale, we used morph colouration to quantify divergence between populations, as populations with greater interbreeding should have more similar colouration. From a standardized dataset containing 4,594 images, those that were low resolution, had poor exposure, were taken from a far distance, or had an object obstructing the webworm were excluded (S3D Fig), leaving 3,789 images. The colour data for the entire fall webworm (including head capsule and body) was extracted using Adobe Photoshop 2021. Individual larvae were selected from each image using Photoshop's object detection tool (S3A Fig). If Photoshop's object detection tool was unable to properly select the webworm, the freehand lasso tool was used instead (S3B and S3C Fig). These colour records were further filtered to those from 2018–2020, east of -100˚W longitude, and in plant hardiness zones 5–9, leaving 3,475 photos for analysis.

To test whether colour data could predict colour morph, we measured the mean, standard deviation, minimum value, maximum value, kurtosis, and skewness of each RGB histogram using image attributes. Colour histograms of images were generated in Photoshop and histogram data was extracted from webworm selections as a.csv file using Photoshop's "Record Measurement" feature (S4 Fig). Using R, we calculated photograph properties manually from exported histogram data. These variables were then normalised (i.e., centered and scaled; [57])

in R in preparation for linear discriminant analysis (LDA) [58]. LDA was performed to find a single variable (LD1) that best separated the red and black colour morphs based on the RGB histogram data. LD1 loadings of colour histogram variables inputted to the LDA are available in S1 Table. To determine if the colour (LD1) of each colour morph depends on plant hardiness zone and year, we performed an ANCOVA. We also performed two simple linear regressions to determine the effect of plant hardiness zone and year on the colour of each colour morph individually. If divergence between morphs increases with increasing temporal niche space, we expect the colour phenology of the morphs to diverge in larger plant hardiness zones. If divergence increases at higher latitudes, we expect colour phenology of the morphs to diverge in smaller plant hardiness zones.

## Results

### Differences between geographic distribution of morphs

We found that red and black morph larvae occurred together across much of their Eastern range (Fig 1 and S1 Fig), consistent with a current sympatric mode of speciation. Larvae in Colorado and northwestern North America were completely geographically isolated from Eastern populations (Fig 1).

While the two morphs are generally found sympatrically, differential abiotic tolerance may change colour morph distribution within their eastern range (east of -100˚W longitude). Black morph populations were found an average of 2.3 ± 1.4 (SD) degrees further north than the red morph populations ($F_{1,5170}$ = 93.00, $P$<0.0001; Fig 2). However, differences in latitudinal range differed by year ($F_{4,5170}$ = 39.12, P<0.0001), as the average ranges of the two morphs did not differ in 2020 (Tukey Post-Hoc, $P$ = 0.06). We also found that red morph populations occurred an average of 4.1 ± 1.5 (SD) degrees further west than the black morphs ($F_{1,5170}$ = 104.30, $P$<0.0001; Fig 2). However, the difference between the longitudinal ranges of morphs also varied among years ($F_{4,5170}$ = 11.87, $P$<0.0001), as although the average longitudinal distributions of morphs in 2020 were significantly different, the magnitude of the difference in longitude was smaller than in previous years (Tukey HSD, $P$<0.0001). There was no difference in the elevation of the red and black colour morphs across all years ($F_{1,4644}$ = 0.06, $P$ = 0.8; Fig 2).

### Differences between phenology of morphs across plant hardiness zones

To find evidence for a difference in phenology between morphs across latitude (Fig 3), we tested for differences in each plant hardiness zone and year between the earliest 10% of observation dates of each morph, and the latest 10% of observation dates of each morph. Previous studies show the black morph appears earlier than the red and remains later in the fall [6]. Overall, we found the earliest 10% of black morph observations occurred an average of 22 ± 3.35 (standard error of mean difference) days earlier than the earliest 10% of red morph observations ($F_{1,521}$ = 156.7, $P$<0.0001; Fig 4A). We found differences between the earliest 10% of morph dates differed by plant hardiness zone. Black morph observations tended to appear earlier across increasing plant hardiness zone (general linear model, β = -23 days/zone) than the red morph (general linear model, β = -18 days/zone), resulting in greater differences between morphs in warmer plant hardiness zones at lower latitudes ($F_{1,521}$ = 876.8, $P$<0.0001; Fig 4A). Differences between earliest observance dates of the morphs across plant hardiness zones were not driven by year ($F_{2,521}$ = 1.81, $P$ = 0.1; Fig 4A). We found the latest 10% of black morph larvae occurred an average of 3 ± 2.27 (standard error of the mean difference) days later than the red morph; however, this difference was not statistically significant ($F_{1,517}$ = 3.77, $P$ = 0.05; Fig 4B). Differences between the 10% of latest dates in the black and red morph were driven by plant hardiness zone. Black morph observations tended to appear later across

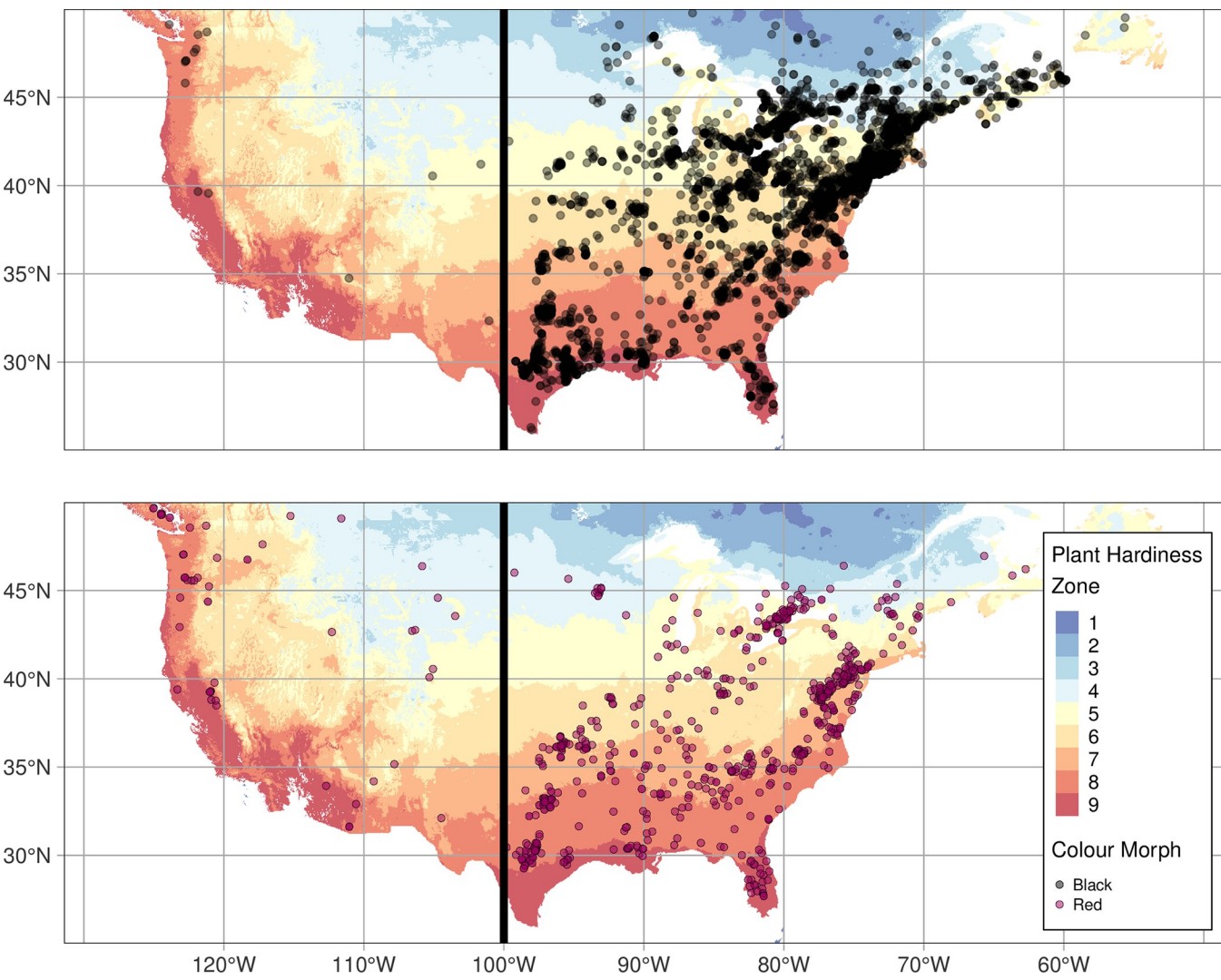

**Fig 1. Distribution of red and black morph fall webworm observations on iNaturalist from 2018–2020 in Canada and the United States.** Each point corresponds to a single iNaturalist observation. Unofficial USDA plant hardiness zones indicated on map to show differences in geographic distribution among morphs. Only observations east of -100˚W longitude (indicated by black line) were used in further analysis. A smaller value of plant hardiness zone correlates to colder climates existing at higher latitudes. n = 4472 for black morph, n = 775 for red morph. Distribution of plant hardiness zones in the United States is derived from the USDA plant hardiness zone shapefiles (https://planthardiness.ars.usda.gov/, [45]), but is not the official USDA Plant Hardiness Zone Map as it includes Canadian plant hardiness zones classified using the same climate bins as the USDA, generated using the BioSIM stochastic weather generator [47].

increasing plant hardiness zones (general linear model, β = 17 days/zone) than the red morph (general linear model, β = 10 days/zone; $F_{2,517}$ = 410.2, $P$<0.0001; Fig 4B); however, since the latest observed dates of the morphs converged in plant hardiness zone 7, there is no trend in the difference between morphs across latitudes. Differences between morph latest observation dates across plant hardiness zones did not differ by year ($F_{2,517}$ = 0.74, $P$ = 0.6; Fig 4B).

We also tested for differences between phenology of morphs by comparing their temporal abundance patterns in each year. Using the DTS statistic, we quantified the difference between the cumulative distribution functions of the morphs across each plant hardiness zone and year (see S2 Fig for cumulative distribution functions of morphs). The relationship between differences in morph temporal abundance and plant hardiness zone was best fit by a linear regression according to AIC [56]. The regression showed differences in the temporal abundance of

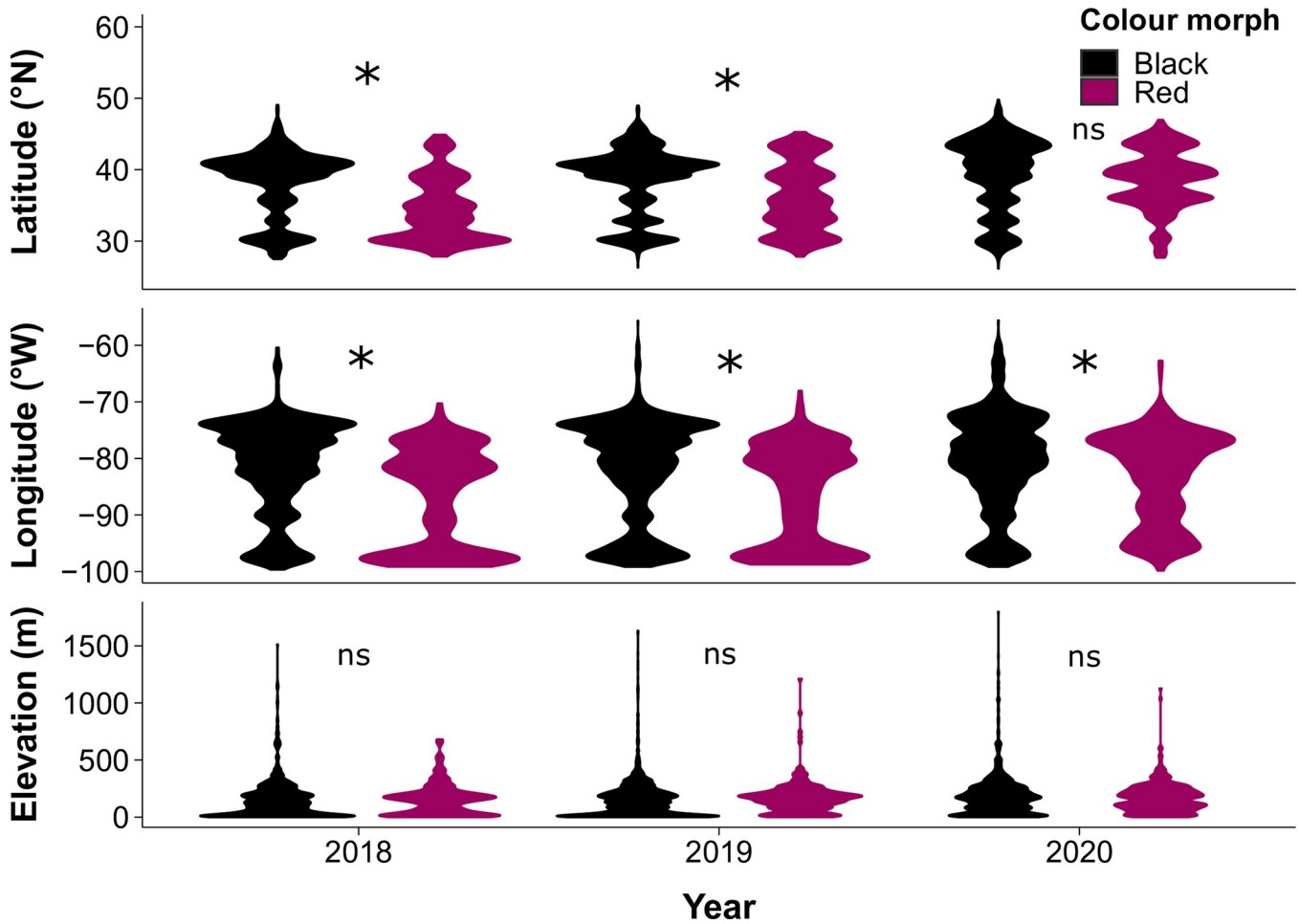

**Fig 2. The red and black fall webworm colour morphs have different distributions across latitude and longitude but not elevation in their eastern North American range (east of -100˚W longitude).** Violin plots indicate scaled density of observations at each value. Asterisks indicate groups are significantly different according to Tukey Post-Hoc test, and "ns" indicates no significance. The geographical range of the black morph averages 2.3 degrees further north (P<0.0001) and 4.1 degrees further east (P<0.0001) than the red morph, but there was no difference in average elevation between the two colour morphs (P = 0.8). n = 5176 for latitude and longitude, n = 4652 for elevation.

the morphs increased with increasing plant hardiness zones ($F_{1,9}$ = 25.11, P<0.001; Fig 5). Trends in temporal overlap across plant hardiness zones did not differ by year ($F_{2,9}$ = 0.15, P = 0.9; Fig 5). DTS analysis was also run on all observations with and without head capsule visible, and no qualitative differences in results were observed.

### Differences in colour morphology of morphs across plant hardiness zones

If there are differences in the effectiveness of allochronic speciation across plant hardiness zones, we would expect to see differences in the separation of colour morphology across plant hardiness zones. Of photos used in colour analysis, 3082 were of black morph webworms while 393 were of red morph webworms. To test the relationship between colour and latitude, we used an LDA model to create a single summary variable (LD1) that best separated the red and black morphs. Among the coefficients of linear discriminants, the mean green value, mean red value, and red standard deviation had the highest absolute values (-2.60, 1.75, and -0.97 respectively; S1 Table), indicating they were most important in distinguishing red and black morph colouration.

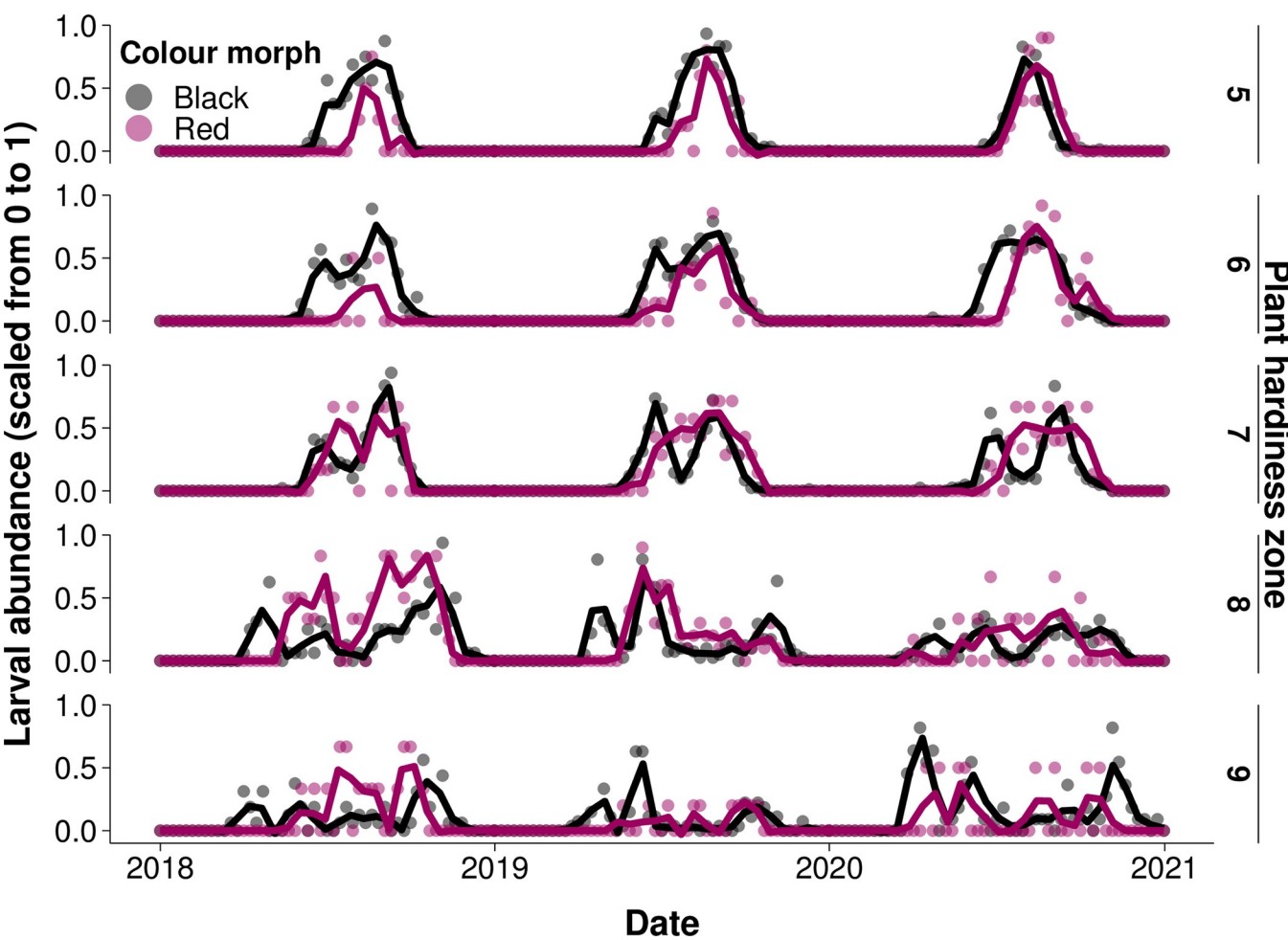

**Fig 3. Phenology of red and black morph fall webworm across years and plant hardiness zones.** Points indicate scaled larval abundance in each week of the year, while lines show trends in abundance across years calculated using the "loess" function.

We found that the colour morphs were sorted well by linear discriminant analysis of their colouration ($F_{1,3467}$ = 925.97, $P<0.0001$; Fig 6). We also found that trends in colouration across plant hardiness zones differed between red and black morphs (interaction between colour morph and plant hardiness zone, $F_{1,3467}$ = 7.37, $P<0.01$). When modelling only red morph colouration, the red larvae show a slight increase in LD1 values as plant hardiness zone increases (linear regression, $\beta$ = 0.1, $R^2$ = 0.004; Fig 6); however, this slope was not significantly different from 0 ($F_{1,387}$ = 0.39, $P$ = 0.5). Year of observation did not affect trends in red morph colouration across plant hardiness zones ($F_{2,387}$ = 1.20, $P$ = 0.3). However, LD1 values decrease slightly with increasing plant hardiness zone when modelling only black morph larvae (linear regression, $\beta$ = -0.09, $R^2$ = 0.016; $F_{1,3078}$ = 42.57, $P<0.0001$; Fig 6). Trends in black morph colouration across plant hardiness zones were also driven by year, with 2019 showing greater decline in LD1 values than other years ($F_{2,3078}$ = 3.7, $P<0.05$).

## Discussion

Here we show that the black morph emerges from overwintering earlier than the red morph in plant hardiness zones which occur at lower latitudes (Fig 4A). We also find that differences

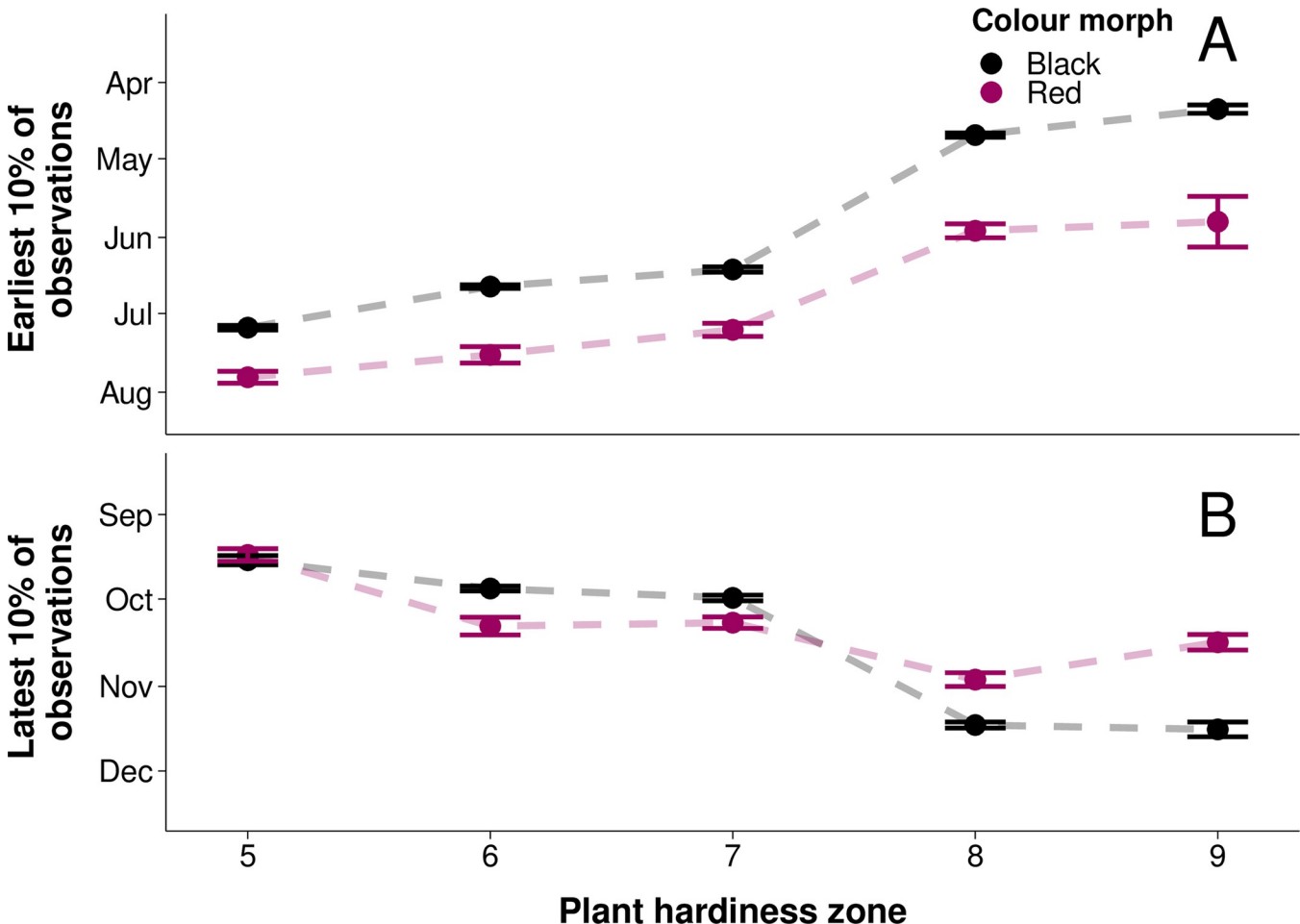

**Fig 4. Difference between mean earliest 10% and latest 10% of observation dates between red and black morphs across plant hardiness zones.** Points show mean of earliest or latest dates, and error bars show standard error of the mean. **A)** The earliest 10% of black morph observations occurred an average of 22 ± 3.35 (standard error of the mean difference) days earlier than the red morph (P<0.0001). Differences between mean dates of the morphs increased in larger plant hardiness zones (P<0.0001). n = 529. **B)** There was no significant difference between the average latest observance date of the red and black morphs (P = 0.05). The black morph tended to occur later than the red in increasing plant hardiness zone (P<0.0001); however, there were no trends in the differences between morphs across zones. n = 525.

between the phenology of the morphs across their entire reproductive season are greater in plant hardiness zones at lower latitudes where the length of the reproductive season is longer (Fig 5). The colouration of the black morph also diverges more from the red morph in plant hardiness zones at lower latitudes (Fig 6). These results suggest that greater differences in phenology at lower latitudes where temporal niche space is larger may be facilitating decreased gene flow between morphs, and that the strength of allochronic speciation varies along a latitudinal gradient.

While the colour morphs of reproductive adult fall webworms cannot be reliably differentiated [6], differences between the emergence dates of red and black morph fall webworm larvae suggests there is greater temporal separation of the morphs at lower latitudes. We found that differences between the earliest observation dates of colour morph larvae were larger in warmer plant hardiness zones at lower latitudes (Fig 4A). There was no difference in the latest observation dates of the morphs, nor was there a consistent trend in differences across plant hardiness zones (Fig 4B). Since the fall webworm overwinters as a pupa and emerges as a

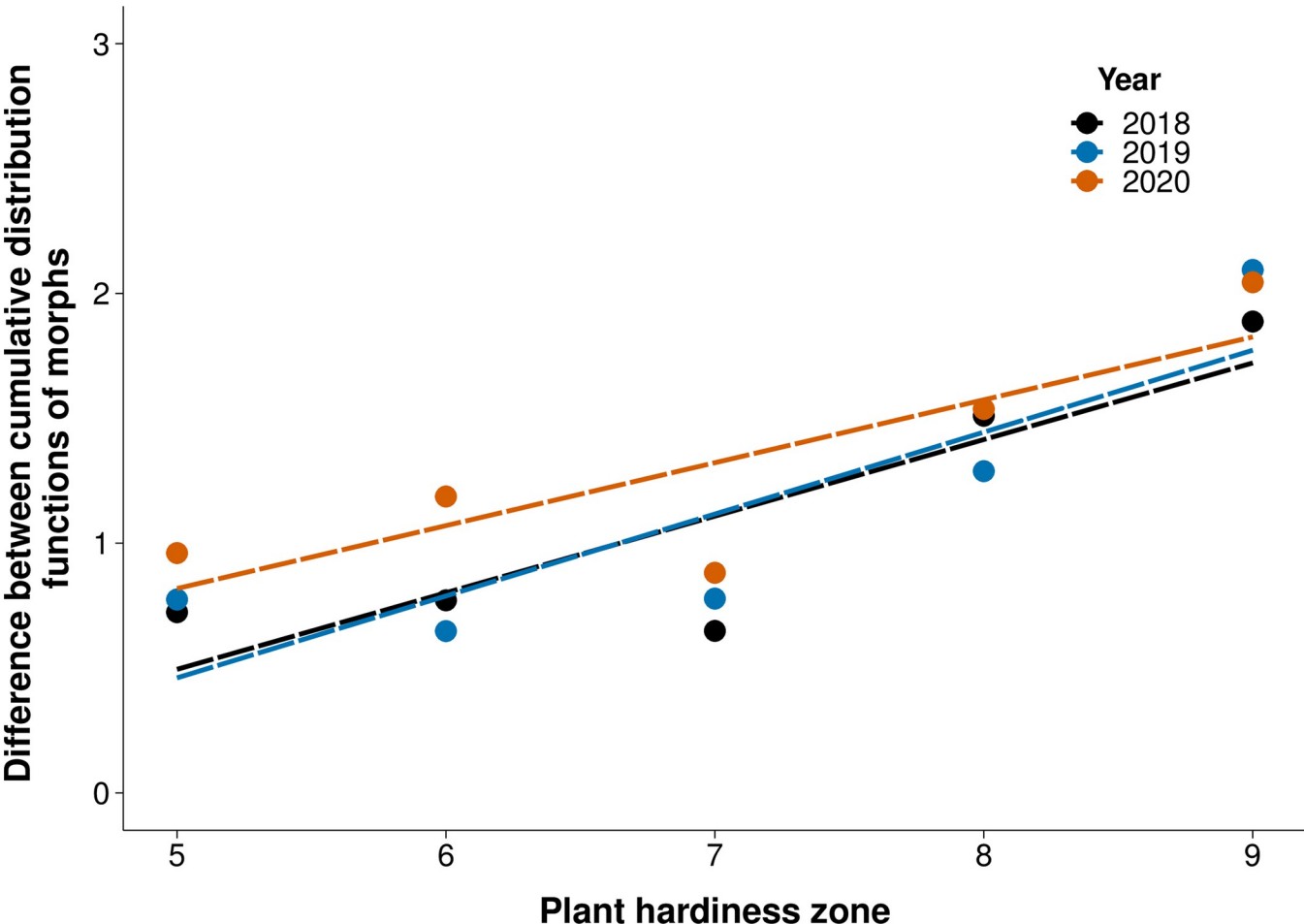

**Fig 5. Differences between cumulative distribution functions of red and black colour morphs increase at warmer plant hardiness zones.** Points indicate value of DTS function (overlap between CDFs) of colour morph populations at each plant hardiness zone and year, lines indicate linear regression of points in each year. Differences between temporal abundance of the morphs increased at warmer plant hardiness zones (P<0.001). n = 15.

reproductive adult [59], differences between the emergence timing of colour morph moths may decrease mating between morphs in the first generation [6]. As well, differences in emergence date between morphs were consistent with differences in diapause intensity found in previous studies of fall webworm [6]. Differences between red and black morph diapause intensity may lead to hybrids between morphs being poorly temporally adapted to their environment, reinforcing genetic separation [6, 60]. These results support our hypothesis that greater temporal niche space at lower latitudes, not winter seasonality and diapause effects at higher latitudes, drives a latitudinal gradient in allochronic speciation. Larger temporal niche space at lower latitudes allows for greater temporal niche partitioning between populations, including changes in seasonal onset and decline. This phenomenon may be facilitating greater differences in emergence date of the colour morphs, and changing the strength of allochronic speciation across latitudes [21].

The phenology of fall webworm larval colour morphs diverged more in warmer climates, suggesting allochronic speciation driven by a difference in phenology operates more effectively at lower latitudes. Evaluating the yearly phenology of the colour morphs was important to determine whether temporal isolation between morphs is maintained across the entire reproductive season [6]. While we cannot differentiate between allochronic speciation being the

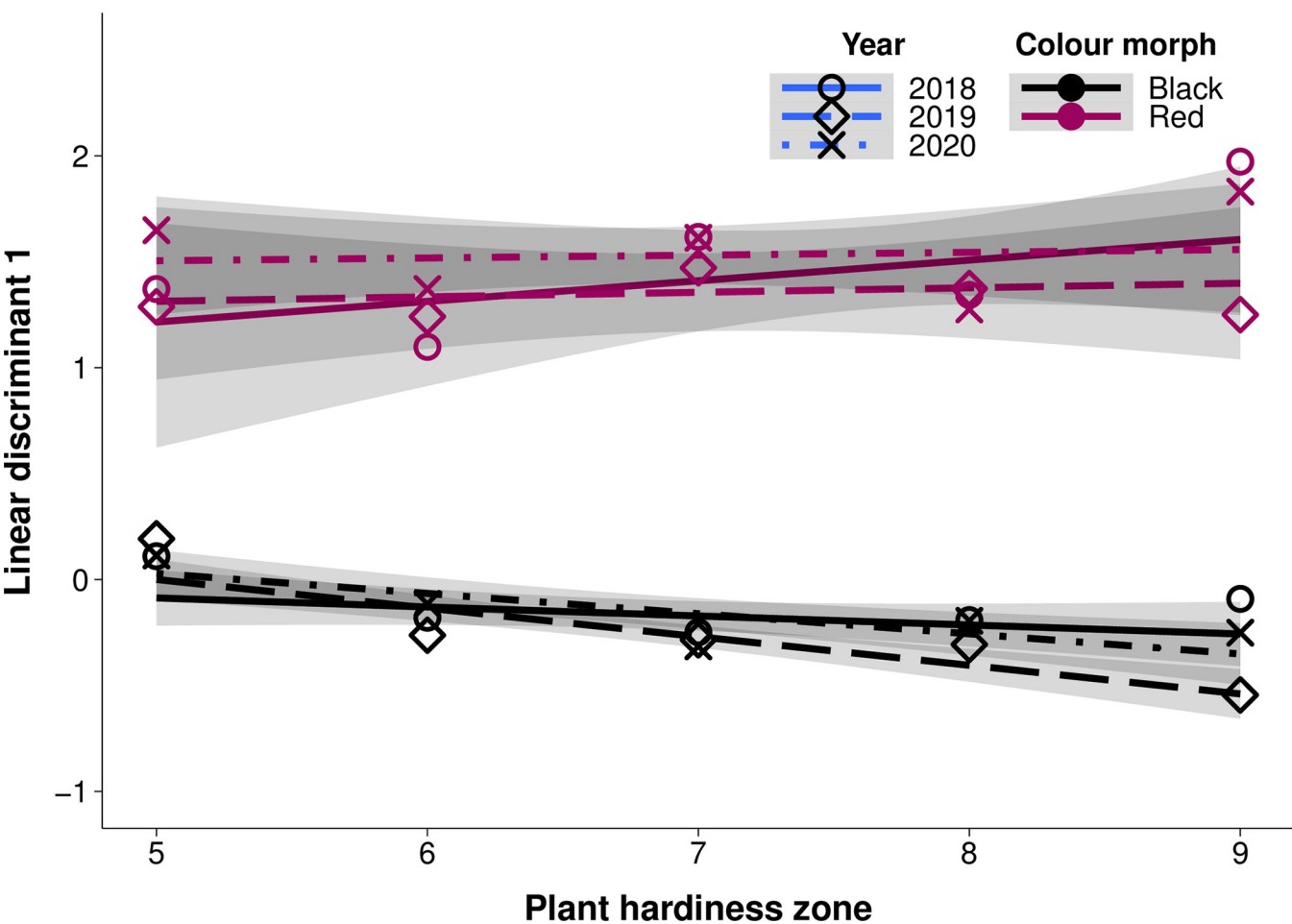

**Fig 6. Colouration of black and red morph fall webworm larvae diverges more at warmer plant hardiness zones.** Line type indicates year, while line colour indicates larval colour morph. Points indicate mean LD1 values of morphs in each plant hardiness zone and year. Grey shaded region shows 95% confidence interval of each year and morph. Red morph LD1 values do not change across zones (P = 0.5) while black morph LD1 values decrease at warmer plant hardiness zones (P<0.0001). n = 3475.

primary cause of divergence or maintaining separation of the morphs after a period of allopatry [6, 33], differences in phenology between morphs are likely driving current trends in divergence and influencing levels of interbreeding between populations across their extensive sympatric range. We found that temporal patterns of abundance of the colour morphs were less similar in climates at lower latitudes (Fig 5). Qualitatively, we also observed fall webworm voltinism was more similar at higher latitudes (Fig 3); however, we could not reliably measure the number of fall webworm generations using these data. These patterns may be driven by trends in temporal niche space, as larger temporal niche space in warmer climates can facilitate greater niche partitioning [21]. The importance of generational overlap on the progress of allochronic speciation across a broad range has been demonstrated in other insects [15, 60]. For example, populations of the geometrid winter moth (*Inurois sp.*) undergoing allochronic speciation showed increased genetic differentiation when population phenology overlapped less, which was facilitated by a difference in seasonality [61]. While our phenological results support a latitudinal gradient in the strength of allochronic speciation driven by temporal niche space, genetic studies on the relatedness of the morphs across a latitudinal range are needed to lend further support to this hypothesis.

Since colour morph is only reliably discernable from larval observations [6], we could not model the phenology of the reproductive form (the moth). Using larval phenology to predict allochronic speciation assumes that both the red and black morph moths appear in the same fixed interval after their larval occurrence. Timing of the life cycle stages of the morphs may have diverged, causing moth phenology to disagree with the larval phenology presented here. Fig 3 is consistent with phenology modelling of moths presented by Yang et al. [6], except at the highest latitude regions where they find univoltine black and red morph moths do not have overlapping generations, suggesting greater differences in phenology than observed here (Fig 5). The discrepancy between our results and those of Yang et al. [6] at the highest latitudes should be explored, as our interpretation relies on larval occurrence being a reliable indicator of moth phenology.

Crowd-sourced data from iNaturalist lent itself well to our broad-scale analysis of fall webworm phenology and phenotype, as it allowed us to investigate macroecological trends across a large geographic and temporal scale [36, 62]. While the lack of consistency in sampling methods of iNaturalist images may have caused increased variation in the data, the volume and spread of data available allowed us to explore questions which cannot be answered using field studies alone. Crowd-sourced photographs may be biased towards days and seasons with favourable weather conditions and urban environments with high observer density, and poor image quality may make colour morph identification more difficult [53]. To address this, images were manually reviewed for species and colour morph, only images in which the head capsule was clearly visible were used, and we calculated larval abundance and phenology by summarizing observations weekly. As well, biases towards observations in urban areas should not affect our results since our comparisons between sympatric morphs do not depend on observer density. However, the accuracy of our phenology and phenotype measures may still have been affected, and investigations using a single image or small subset of observations from crowd-sourced datasets such as the one used here may be unreliable. In future studies using crowd-sourced science, consideration should be given to how to minimize confounding factors due to uneven sampling, such as using crowd-sourced science data in tandem with field surveys [38]. However, the accessibility of iNaturalist data, ease of collection, volume of data available, and usefulness of crowd-sourced data for broad-scale research makes it a promising avenue for data collection, and a useful resource for investigating macroecological trends across latitudes.

Phenotypic results showed colouration of red and black morph caterpillars diverged in warmer climates. The colour phenotype of the black morph appears to shift away from the colour phenotype of the red morph at warmer plant hardiness zones. Interestingly, this shift in colour only appears in the black morph, as red morph webworms do not experience any significant colour shift across their entire range (Fig 6). If the colour shift were a result of increasing melanism in higher latitude populations, as observed in other ectotherms [63], we would expect to observe this shift in both colour morphs. However, changes in colouration across the fall webworm range were small, and qualitatively we observed variation in colour phenotype even among individuals of the same region. So, these results lend some support to our hypothesis that the strength of allochronic speciation is higher at lower latitudes, but further investigation using field and genetic studies is needed.

While a genetic comparison of colour morphs across latitudes could determine the degree speciation between morphs and support phenotypic results presented here, a genetic dataset on a geographic scale large enough to address this question does not exist [6]. We attempted to quantify the genetic relatedness of the morphs across their range, but available data was not sufficient to answer this question (unpublished data). Morphological and phenotypic comparisons such as the ones performed here can offer valuable information on the relatedness of

populations undergoing speciation prior to investigating population genetic structure. For example, the bean leaf beetle (*Cerotoma trifurcate*) has four distinct colour phenotypes with a previously unknown evolutionary relationship. Morphological analyses of the colour phenotypes were used to generate an initial hypothesis regarding their phylogenetic separation, and genetic work confirmed weak reproductive separation between phenotypes [64]. Trends in the colour morphology and phenology of the fall webworm across their latitudinal range inform hypotheses on the strength of allochronic speciation on a latitudinal gradient; however, we encourage further investigation into population genetics of the fall webworm to confirm the results presented here.

## Conclusion

In this study we tested two competing hypotheses: the first that allochronic speciation facilitates divergence between populations more effectively at higher latitudes, or the second that allochronic speciation facilitates divergence between populations more effectively at lower latitudes, using the fall webworm (*Hyphantria cunea*) as a model species. Our analyses of phenology and phenotypic differences are consistent with the current allochronic speciation in the fall webworm operating more effectively at lower latitudes where there is greater temporal niche space, rather than at higher latitudes due to the effects of winter seasonality and differences in diapause. Further study is needed as some factors such as limited access to genetic data, variable sampling methods in crowd-sourced science data, and using only larval phenology measures may have confounded our results. Advances in macroecology and crowd-sourced data provide an avenue for further study of ecological speciation over large geographic areas. Furthermore, investigating the interplay of latitudinal effects and ecological speciation can change our understanding of how speciation operates under different conditions and help to understand past, present, and future trends in biodiversity. Since the great diversity of phytophagous insects has been attributed to high rates of sympatric speciation, greater exploration of allochronic speciation and other forms of ecological speciation in widespread species could elucidate this evolutionary history [16, 17].

## Supporting information

**S1 Table. Coefficients of linear discriminants from LDA analysis of variables separating colouration of red and black morphs in iNaturalist photographs.** Among the coefficients of linear discriminants, the mean green value, mean red value, and red standard deviation have the highest absolute values (-2.60, 1.75, and -0.97 respectively).
(DOCX)

**S1 Fig. Regional distribution of iNaturalist fall webworm observations from 2018–2020 with colour morph indicated.** Red and black morph fall webworms occur sympatrically at a regional and local level, implicating a sympatric form of speciation. Regions shown are southern Ontario, northeastern USA, and southern Texas. All maps generated using Simplemappr (https://www.simplemappr.net/), for which maps are available in the public domain without license.
(PDF)

**S2 Fig. Cumulative distribution functions of red and black observations in each year and plant hardiness zone.** Points indicate cumulative proportion of observations, while lines show trends in distribution across years.
(PDF)

**S3 Fig. Fall webworm image isolation and selection in Adobe Photoshop for colour phenotype analysis.** A) Complete isolation of fall webworm larva by Photoshop using object detection tool, no manual adjustment needed. B) Incomplete selection of fall webworm larva by object detection tool. Corrected manually by researcher using freehand lasso tool. C) Complete selection of fall webworm larva in Photoshop after manual correction using freehand lasso tool. D) Example of photo removed from colour phenotype analysis. Photo was low resolution, had poor exposure, was taken from a far distance, and had an object obstructing the webworms. Photos shown here obtained from iNaturalist.org under public domain licensing (CC0).
(PNG)

**S4 Fig. Animated RGB histogram of red and black fall webworm larvae across their latitudinal range.**
(GIF)

## Acknowledgments

We humbly thank J. Stireman at the University of Denver for his preliminary review of our results, and Dr. S. Otto for helpful comments that significantly improved the manuscript. We also thank each community member who submitted their photo to iNaturalist to aid in our data collection. We are appreciative of our friends and family who supported us throughout this project.

## Author Contributions

**Conceptualization:** Emily N. Black, Katie E. Marshall.

**Data curation:** Emily N. Black, Jarrett D. Blair, Karin R. L. van der Burg, Katie E. Marshall.

**Formal analysis:** Emily N. Black, Jarrett D. Blair, Karin R. L. van der Burg, Katie E. Marshall.

**Funding acquisition:** Katie E. Marshall.

**Investigation:** Emily N. Black, Jarrett D. Blair, Katie E. Marshall.

**Methodology:** Emily N. Black, Jarrett D. Blair, Katie E. Marshall.

**Project administration:** Katie E. Marshall.

**Resources:** Katie E. Marshall.

**Supervision:** Katie E. Marshall.

**Visualization:** Emily N. Black, Jarrett D. Blair, Katie E. Marshall.

**Writing – original draft:** Emily N. Black, Jarrett D. Blair, Karin R. L. van der Burg, Katie E. Marshall.

**Writing – review & editing:** Emily N. Black, Jarrett D. Blair, Karin R. L. van der Burg, Katie E. Marshall.

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
