## [Decision Letter · Decision Letter 0]

4 May 2023

PONE-D-23-05021Crowd-sourced observations of a polyphagous moth reveal evidence of allochronic speciation varying along a latitudinal gradientPLOS ONE

Dear Dr. Black,

Thank you for submitting your manuscript to PLOS ONE. After careful consideration, we feel that it has merit but does not fully meet PLOS ONE’s publication criteria as it currently stands. Therefore, we invite you to submit a revised version of the manuscript that addresses the points raised during the review process.

I am providing my own review.

The authors present fall webworm larval occurrence data from iNaturalist to test competing hypotheses about factors influencing the strength of allochrony. The study is presented in the context of understanding whether available phenological niche space or intensity of seasonality are the primary drivers. The hypotheses and logic of the study are clear and the authors have accumulated a great deal of data from citizen scientists to test there hypotheses. I found the manuscript to be compelling and well-written. While the authors rightly suggest that molecular genetics data are needed to fully test their ideas, the phenological data presented here support their interpretations and motivate fuller testing with population genetics data.

One question arises from the authors' contention (assumption?) that the evolutionary history of H. cunea is largely one of allochronic speciation. This is a conclusion that Yang et al. (2017) arrived at. Perhaps I am not familiar with all of the evidence, but I wonder if this is fully supported because the evidence for this seems rather weak since the two forms are quite diverged, at least as measured by mtDNA data of Yang et al.. The alternatives might include a period of allopatry prior to secondary contact/sympatry (via Pleistocene climate change or range expansions since then?). In this scenario, allochrony, as the authors' see it today, might maintain isolation in sympatry, but not be the primary cause of divergence. In other words, the language of allochronic speciation in the introduction might be examined and augmented with a fuller detailing of the evidence. 

Below are some minor comments.

line 296 change to "these data" or "this data set"

lines 452, 454: this is unclear: "will increase in larger plant hardiness zones." Perhaps readers could be reminded that zones with higher numbers correspond to lower latitudes/longer growing seasons.

line 495: one variable in LDA? "find a variable that best separated the red and black colour morphs based on the RGB histogram". Inspection of supplementary Table 1 indicates that this is in fact a multivariate discriminant function.

We look forward to receiving your revised manuscript.

Kind regards,

Christopher Nice, Ph.D.

Academic Editor

PLOS ONE

Journal Requirements:

3. We note that Figure 1 & Supporting Figure 1 in your submission contain [map/satellite] images which may be copyrighted. All PLOS content is published under the Creative Commons Attribution License (CC BY 4.0), which means that the manuscript, images, and Supporting Information files will be freely available online, and any third party is permitted to access, download, copy, distribute, and use these materials in any way, even commercially, with proper attribution. For these reasons, we cannot publish previously copyrighted maps or satellite images created using proprietary data, such as Google software (Google Maps, Street View, and Earth). For more information, see our copyright guidelines: http://journals.plos.org/plosone/s/licenses-and-copyright.

 a. You may seek permission from the original copyright holder of Figure 1 & Supporting Figure 1 to publish the content specifically under the CC BY 4.0 license. 

Reviewers' comments:

Reviewer's Responses to Questions

**Comments to the Author**

1. Is the manuscript technically sound, and do the data support the conclusions?

Reviewer #1: Partly

2. Has the statistical analysis been performed appropriately and rigorously? 

Reviewer #1: Yes

3. Have the authors made all data underlying the findings in their manuscript fully available?

Reviewer #1: Yes

4. Is the manuscript presented in an intelligible fashion and written in standard English?

Reviewer #1: Yes

5. Review Comments to the Author

Reviewer #1: The crowd-based data seems to be a powerful tool to analyze life history as the current investigation demonstrated but the identification of the Northwestern black is wrong. Yang et al. 2017 described it "maloon type" and Hattori and Ito described it the "mottled" though the phenotype looks like black if we only depend on the head color. However, larval behavior, web type and coat color are not exactly the same as Eastern black and more close to the eastern red, though the west Washington populations have bright orange head capsules. Yan et al. demonstrated the Northwestern populations enter deep diapause distinctly from the black from the eastern states. If the authors insist that they are realy the black, alochronic speciation occurred indipendently at the both sides of the rocky mountains?

Two sets did not undergo the allopatric speciation?

6. PLOS authors have the option to publish the peer review history of their article (what does this mean?). If published, this will include your full peer review and any attached files.

Reviewer #1: **Yes: **xx

---

## [Author Response · Author response to Decision Letter 0]

7 Jun 2023

Response to Reviewers

*Please note that for ease of reading, and since the manuscript no longer needs to follow PLOS Biology organizational standards, the methods have been moved to immediately after the introduction. This has changed line references from original reviewer comments; however, changes should be easily visible in the marked-up word document uploaded in the file package, and all updated line numbers corresponding to the marked-up word document are indicated here.*

Reviewer 1 (Editor): 

The authors present fall webworm larval occurrence data from iNaturalist to test competing hypotheses about factors influencing the strength of allochrony. The study is presented in the context of understanding whether available phenological niche space or intensity of seasonality are the primary drivers. The hypotheses and logic of the study are clear and the authors have accumulated a great deal of data from citizen scientists to test there hypotheses. I found the manuscript to be compelling and well-written. While the authors rightly suggest that molecular genetics data are needed to fully test their ideas, the phenological data presented here support their interpretations and motivate fuller testing with population genetics data.

We thank the editor for their kind words and appreciate the time they have taken to review our manuscript. 

One question arises from the authors' contention (assumption?) that the evolutionary history of H. cunea is largely one of allochronic speciation. This is a conclusion that Yang et al. (2017) arrived at. Perhaps I am not familiar with all of the evidence, but I wonder if this is fully supported because the evidence for this seems rather weak since the two forms are quite diverged, at least as measured by mtDNA data of Yang et al.. The alternatives might include a period of allopatry prior to secondary contact/sympatry (via Pleistocene climate change or range expansions since then?). In this scenario, allochrony, as the authors' see it today, might maintain isolation in sympatry, but not be the primary cause of divergence. In other words, the language of allochronic speciation in the introduction might be examined and augmented with a fuller detailing of the evidence. 

We appreciate this feedback and agree that a more nuanced discussion of potential allopatry in this species is needed. We also thank the editor for their detailed review of the literature to guide our changes. Upon further investigation of the literature, we did not find strong evidence to suggest allopatry facilitated historical divergence of this group; however, we also did not find strong evidence to the contrary. Thus, further discussion of this is warranted in the introduction and discussion. 

We have included more general statements in the introduction detailing the interactions of allochrony with other types of speciation: 

Line 56: “Allochronic speciation may interact with others forms of speciation; for example, it may maintain genetic isolation of formerly allopatric populations experiencing secondary contact [6,7], evolve more easily in populations whose phenotypes are already partially diverged [8,9], or occur in tandem with other types of ecological speciation, such as host plant specialization [10].”

Line 62: “This non-overlap of host plant phenology imposed a temporal reproductive barrier between R. pomonella host plant races, leading to allochronic speciation (however, this speciation may have been facilitated by allopatry; see [8,9]) [10].”

Please note that several new references were added above to support our discussion of the interaction of allopatry and allochronic speciation. 

We also discuss the role of allopatry in separation of the fall webworm in the introduction: 

Line 138: “The initial divergence of the two populations was traced to 1.2-1.6 Mya, suggesting the two morphs may have initially diverged due to allopatric speciation as a result of historical glaciation or other geographic barriers [6,33]. However, the ability of current populations to produce viable offspring when bred experimentally yet remain distinct in the wild suggests the existence of a barrier maintaining separation of the morphs.” 

Please note an additional reference has been added – Masaki & Ito, 1997. This is the source of Yang et al. 2017’s discussion of allopatry in the fall webworm. 

We also made changes to wording throughout the introduction to ensure consistency of these ideas. 

In the discussion, we added the following in our discussion of differences in phenology: 

Line 560: “While we cannot differentiate between allochronic speciation being the primary cause of divergence or maintaining separation of the morphs after a period of allopatry [6,33], differences in phenology between morphs are likely driving current trends in divergence and influencing levels of interbreeding between populations across their extensive sympatric range.”

line 296 change to "these data" or "this data set"

Completed as requested at line 567. 

lines 452, 454: this is unclear: "will increase in larger plant hardiness zones." Perhaps readers could be reminded that zones with higher numbers correspond to lower latitudes/longer growing seasons.

Completed as requested, now reads: 

Line 298: “…we expect differences between the earliest and/or latest observation dates of the morphs will increase in higher plant hardiness zones at lower latitudes with longer growing seasons. Alternatively, if divergence between the morphs increases at higher latitudes, we expect differences between observation dates of the morphs will decrease in higher plant hardiness zones at lower latitudes.”

line 495: one variable in LDA? "find a variable that best separated the red and black colour morphs based on the RGB histogram". Inspection of supplementary Table 1 indicates that this is in fact a multivariate discriminant function.

While the analysis performed was true linear discriminant analysis with a single output variable, we note that the reference to Table S1 in the sentence as written was misleading, and we thank the reviewer for their feedback. We have adjusted it to read: 

Line 362: “LDA was performed to find a single variable (LD1) that best separated the red and black colour morphs based on the RGB histogram data. LD1 loadings of colour histogram variables inputted to the LDA are available in S1 Table. “

Reviewer 2: 

The crowd-based data seems to be a powerful tool to analyze life history as the current investigation demonstrated but the identification of the Northwestern black is wrong. Yang et al. 2017 described it "maloon type" and Hattori and Ito described it the "mottled" though the phenotype looks like black if we only depend on the head color. However, larval behavior, web type and coat color are not exactly the same as Eastern black and more close to the eastern red, though the west Washington populations have bright orange head capsules. Yan et al. demonstrated the Northwestern populations enter deep diapause distinctly from the black from the eastern states. If the authors insist that they are realy the black, alochronic speciation occurred indipendently at the both sides of the rocky mountains?

Two sets did not undergo the allopatric speciation?

We thank the reviewer for their time reviewing our manuscript. We appreciate that classification of the maroon morph in the fall webworm’s Northwestern range is complex, and that our understanding of its phylogenetic relatedness to the eastern populations is limited. We also understand that our method of webworm classification was unable to reliably classify the maroon morph. For these reasons, as well as general data scarcity, we excluded fall webworm observations west of -100 ° longitude in our analyses. However, we understand that this could be clearer in Figure 1 and in our discussion of the two morphs. 

We updated the introduction to include this clarification:

Line 121: “Fall webworm larvae occur in two distinct colour morphs in its eastern range: a red morph and a black morph which exist sympatrically [28].” 

We also added a black line to Figure 1 at -100° longitude to indicate the cut-off for observations in our analysis. 

Changes in response to PLOS One publishing guidelines: 

We note that Figure 1 & Supporting Figure 1 in your submission contain [map/satellite] images which may be copyrighted. All PLOS content is published under the Creative Commons Attribution License (CC BY 4.0), which means that the manuscript, images, and Supporting Information files will be freely available online, and any third party is permitted to access, download, copy, distribute, and use these materials in any way, even commercially, with proper attribution. We require you to either (1) present written permission from the copyright holder to publish these figures specifically under the CC BY 4.0 license, or (2) remove the figures from your submission. 

While both maps included in the main figures and supplemental material are freely available for public use and reproduction, the basemap for Fig 1 has specific attribution guidelines. Since modifications were made to the original basemap of Fig 1, as requested by the data providers we included an addendum indicating the basemap is unofficial: 

“Fig 1…Distribution of plant hardiness zones in the United States is derived from the USDA plant hardiness zone shapefiles (https://planthardiness.ars.usda.gov/, [45]), but is not the official USDA Plant Hardiness Zone Map as it includes Canadian plant hardiness zones classified using the same climate bins as the USDA, generated using the BioSIM stochastic weather generator [48].”

We also indicated that Figure S1 is available under public domain licensing:

“Fig S1… All maps generated using Simplemappr (https://www.simplemappr.net/), for which maps are available in the public domain without license.”

---

## [Editor Report · Decision Letter 1]

26 Jun 2023

Crowd-sourced observations of a polyphagous moth reveal evidence of allochronic speciation varying along a latitudinal gradient

PONE-D-23-05021R1

Dear Dr. Black,

We’re pleased to inform you that your manuscript has been judged scientifically suitable for publication and will be formally accepted for publication once it meets all outstanding technical requirements.

Kind regards,

Christopher Nice, Ph.D.

Academic Editor

PLOS ONE
---

## [Editor Report · Acceptance letter]

5 Jul 2023

PONE-D-23-05021R1 

Crowd-sourced observations of a polyphagous moth reveal evidence of allochronic speciation varying along a latitudinal gradient 

Dear Dr. Black:

I'm pleased to inform you that your manuscript has been deemed suitable for publication in PLOS ONE. Congratulations! Your manuscript is now with our production department. 

Kind regards, 

on behalf of

Dr. Christopher Nice 

Academic Editor

PLOS ONE